# Effect of Carbon Sources in Carotenoid Production from *Haloarcula* sp. M1, *Halolamina* sp. M3 and *Halorubrum* sp. M5, Halophilic Archaea Isolated from Sonora Saltern, Mexico

**DOI:** 10.3390/microorganisms9051096

**Published:** 2021-05-20

**Authors:** Ana Sofía Vázquez-Madrigal, Alejandra Barbachano-Torres, Melchor Arellano-Plaza, Manuel Reinhart Kirchmayr, Ilaria Finore, Annarita Poli, Barbara Nicolaus, Susana De la Torre Zavala, Rosa María Camacho-Ruiz

**Affiliations:** 1Biotecnología Industrial, Centro de Investigación y Asistencia en Tecnología y Diseño del Estado de Jalisco, Guadalajara A.C. 44270, Mexico; anasofiavma@gmail.com (A.S.V.-M.); alejandrabt@iteso.mx (A.B.-T.); marellano@ciatej.mx (M.A.-P.); mkirchmayr@ciatej.mx (M.R.K.); 2Istituto di Chimica Biomolecolare, Consiglio Nazionale delle Ricerche, Via Campi Flegrei 34, 80078 Pozzuoli, Italy; ilaria.finore@icb.cnr.it (I.F.); apoli@icb.cnr.it (A.P.); bnicolaus@icb.cnr.it (B.N.); 3Facultad de Ciencias Biológicas, Instituto de Biotecnología, Universidad Autónoma de Nuevo León, Av. Pedro de Alba S/N Ciudad Universitaria, San Nicolás de los Garza C.P. 66455, Mexico; susana.delatorrezv@uanl.edu.mx

**Keywords:** *Haloarcula*, *Halolamina*, *Halorubrum*, halophilic, archaea, bacterioruberin, carotenoid, Sonora, Mexico

## Abstract

The isolation and molecular and chemo-taxonomic identification of seventeen halophilic archaea from the Santa Bárbara saltern, Sonora, México, were performed. Eight strains were selected based on pigmentation. Molecular identification revealed that the strains belonged to the *Haloarcula, Halolamina* and *Halorubrum* genera. Neutral lipids (quinones) were identified in all strains. Glycolipid S-DGD was found only in *Halolamina* sp. strain M3; polar phospholipids 2,3-O-phytanyl-*sn*-glycerol-1-phosphoryl-3-*sn*-glycerol (PG), 2,3-di-O-phytanyl-sn-glycero-1-phospho-3′-sn-glycerol-1′-methyl phosphate (PGP-Me) and sodium salt 1-(3-*sn*-phosphatidyl)-*rac*-glycerol were found in all the strains; and one unidentified glyco-phospholipid in strains M1, M3 and M4. Strains M1, M3 and M5 were selected for further studies based on carotenoid production. The effect of glucose and succinic and glutamic acid on carotenoid production was assessed. In particular, carotenoid production and growth significantly improved in the presence of glucose in strains *Haloarcula* sp. M1 and *Halorubrum* sp. M5 but not in *Halolamina* sp. M3. Glutamic and succinic acid had no effect on carotenoid production, and even was negative for *Halorubrum* sp. M5. Growth was increased by glutamic and succinic acid on *Haloarcula* sp. M1 but not in the other strains. This work describes for first time the presence of halophilic archaea in the Santa Bárbara saltern and highlights the differences in the effect of carbon sources on the growth and carotenoid production of haloarchaea.

## 1. Introduction

Halophilic archaea are mostly aerobic, Gram-negative, chemoorganotrophic microorganisms that grow from 1M NaCl to the point of saturation in a temperature range from 4 to 68 °C [1,2]. They produce carotenoid pigments synthesized through the mevalonate pathway in order to tolerate such extreme conditions [3]. Carotenoids are important bioactive compounds with antioxidant and free radical scavenging properties [4]. They have a great economic impact, especially in the market that seeks to replace synthetic pigments with those of natural origin. Bacterioruberin, the main carotenoid pigment in halophilic archaea, is a 50-carbon structure with 13 double bonds, involved in the protection of oxidative stress, solar radiation [5] and influences the fluidity of cell membrane [6]. Bacterioruberin is not exclusive to the archaea domain, as some bacteria (*Rubrobacter radiotolerans*, *Kocuria rosea*) have been reported as bacterioruberin producers [5]. In particular, this pigment has a great industrial potential due to its high antioxidant capacity, which is considerably greater that β-carotene, astaxanthin and other carotenoids [7,8]. In addition, bacterioruberin already has been tested in animal cells, exhibiting no toxicity in human monocytes and improving the viability in ram sperm [8].

Haloarchaea are the main microorganisms present in hypersaline environments. These habitats contain at least three times the sea salt concentration [9]. The Santa Bárbara saltern (Figure 1) is a salt-producing area in a non-protected zone, classified as a thalassohaline environment (NaCl as the main salt), salted wetland, with a permanent connection to the sea located in the south coast of Sonora, México. Santa Bárbara houses a great diversity of species of halophilic plants, birds and mammals [10], and so far, the evidence of microorganisms in Sonora State include halophilic bacteria [11] and methanogenic archaea [12]; however, until this work, the halophilic archaea phenotype was still unknown in this area.

Archaea species populate extreme environments and their challenging identification and characterization make them organisms in need of constant elucidation. The discovery of new halophilic species and the use of different identification tools provokes a continuously changing classification and promotes the emergence of new taxonomic groups. For example, the *Halorubrum* genus, the largest in the class Halobacteria, has been rearranged several times since it was proposed in 1995 [13].

It has been shown that some carbon sources, such as carbohydrates, amino acids and other compounds, could stimulate microbial growth; nevertheless, information on how these carbon sources affect carotenoid production is scarce. The complexity of the substrate and the metabolic profile of the organism are important factors, as the latter has a high variability in archaea species, sometimes even in species within the same genera [14].

Although complete metabolism and the origin of metabolic routes in haloarchaea are still unknown, it is clear that they have acquired genes from bacterial pathways such as mevalonate through horizontal gene transfer [15]. The origin and understanding of this pathway have not been fully decrypted due to the erratic genetic pattern that codes for the enzymes of this route [16]; the carotenoid pigments in these organisms are synthetized by this pathway, which starts with acetyl-CoA, and starting from this point, it should be verify which compounds could stimulate pigmentation. Some halophilic archaea, such as *Halobacterium*, contain enzymes that allow some amino acids, such as glutamic acid, to be directly incorporated into the tricarboxylic acid cycle (TCA) and obtain greater availability of acetyl-CoA that could be used for the mevalonate pathway [17]. Succinic acid also has been used to improve carotenoid production in the yeast *Xantophyllomyces dendrohous*, entering directly into the TCA cycle and increasing the activity of the enzymes [18].

Another way to obtain acetyl-CoA in species of archaea capable of utilizing carbohydrates is from glucose. Glucose is converted to pyruvate in four steps via the Entner–Doudoroff (ED) pathway, and a carboxyl group is removed and coenzyme A is attached to form acetyl-CoA [19]. Gochnauer et al. used various concentrations of glucose, stimulating the growth and the pigmentation process in *Halobacterium*, and verified that this monosaccharide acted optimally at 2% [20,21].

Thus, the aims of this work are to provide information about the halophilic microorganisms that inhabit Santa Bárbara saltern, search for carotenoid producers, perform their chemotaxonomic and molecular identification and to assess the effect of glucose, succinic and glutamic acid on the production of carotenoid pigments in some of the isolated species.

## 2. Materials and Methods

### 2.1. Organisms and Culture Media

The microorganisms used in this study were isolated from saline mud and salt crust samples collected in November 2017. Samples were collected from 10 different points up to 20 cm deep within an 8.2 km^2^ area of the Santa Bárbara saltern, located in the Municipality of Huatabampo, Sonora, México. Soil temperatures in Santa Bárbara range from 35 to 36 °C during the day and has a soil pH from 7.7 to 8.6. A map of the sampling sites is showed in Figure 1. The environmental samples were stored in refrigerated sterile plastic bags until their analysis.

The samples (1 g) were diluted in 9 mL of 20% sterile saline solution and used to inoculate a solid ATCC2185 medium, pH 8, containing (g/L) MgSO_4_ × 7H_2_O 20, sodium citrate 3, KCl 2, tryptone 5, yeast extract 3, NaCl 233.3, agar 10, and trace elements solution 100 µL. Trace element solution was prepared containing (g/L) ZnSO_4_ × H_2_O 1.32 g, MgSO_4_ × H_2_O 0.34 g, Fe (NH_4_)_2_SO_4_ × 6H_2_O 0.82 g, and CuSO_4_ × 5H_2_O 0.14 g. This culture medium is generally employed for the growth of halophilic archaea [22]. After 1 week of incubation at 37 °C in an aerobic incubation stove with no artificial light, the presence of colonies with a different morphology was revealed and they were purified by restreaking on the same solid medium. The isolates were stained with methyl violet for microscopic observations and cryopreserved at −80 °C in 75% glycerol. All reagents were purchased from Sigma-Aldrich, St. Louis, MO, USA.

### 2.2. DNA Extraction and Phylogenetic Analysis

For the DNA analysis, the cells of all isolates were collected after 4 days of incubation at a temperature of 37 °C via centrifugation at 5000× *g* for 15 min.

The cells were subjected to total genomic DNA extraction and purification by using the Gen Elute^TM^ Plant Genomic DNA Miniprep Kit Protocol (Sigma-Aldrich, USA) according to manufacturer’s specifications, followed by a PCR of the complete 16S rRNA gene using forward D30 (5′-ATTCCGGTTGATCCTGC-3′) and reverse D56 (5′GYTACCTTGTTACGACTT-3′) primers [23] and the Invitrogen Platinum *Taq* DNA Polymerase (ThermoFisher Scientific, Waltham, MA, USA) protocol. The following conditions were used: initial denaturation at 94 °C for 3 min, followed by 30 cycles of 94 °C for 45 s, 53.5 °C for 45 s and 72 °C for 2 min, and a final extension at 72 °C for 10 min. The strains were cloned using a CloneJet PCR Cloning Kit (ThermoFisher Scientific, MA, USA) into *E. coli* calcium competent cells. Consensus sequences were generated with the sequenced fragments using software CLC Workbench 8. A BLASTn was performed with sequences from the GenBank 16S database (Bacteria and archaea) [24].

Reference 16S rRNA gene sequences of the closest haloarchaea were selected from the NCBI-GenBank. Sequences were aligned with ClustalW [25] and trimmed using MEGA V 7.0 [26] obtaining a 453 bp sequence alignment. The optimal substitution model was determined with MEGA for the sequence alignment. A calculated T92+1 model [27] was used to reconstruct a phylogenetic tree with a Maximum Likelihood (ML) algorithm in MEGA V 7.0, with 1000 bootstraps.

### 2.3. Chemo-Taxonomic Identification

The lipid extraction was carried out according to Finore et al. [28] using 2.5 g of freeze-dried cells harvested at the stationary growth phase. Quinones were extracted from freeze-dried cells with *n*-hexane and were purified by TLC on silica gel (0.25 mm; F254, Merck) and eluted with *n*-hexane/ethylacetate (96:4, *v*/*v*). The purified UV-bands from TLC were then analysed by LC/MS on a reverse-phase RP-18 Lichrospher column eluted with *n*-hexane/ethylacetate (99:1, *v*/*v*) with a flow rate of 1.0 mL min^−1^ and identified by electrospray ionization (ESI)-MS and ^1^H-NMR spectrometry. NMR spectra were acquired on a Bruker DPX-300 operating at 300 MHz, using a dual probe. The residual cellular pellet, after *n*-hexane extraction of freeze-dried cells, was subjected to extraction with CHCl_3_/CH_3_OH/H_2_O (65:25:4, by vol.) for polar lipids recovery. The polar lipid extract was analysed by TLC on silica gel (0.25 mm, F254, Merck) eluted in the first dimension with CHCl_3_/CH_3_OH/H_2_O (65:25:4, by vol.) and in the second dimension with CHCl_3_/CH_3_OH/acetic acid/H_2_O (8:12:15:4, by vol.). All polar lipids were detected by spraying the plates with 0.1% (*w*/*v*) Ce(SO_4_)_2_ in 1 M H_2_SO_4_ or with 3% (*w*/*v*) methanolic solution of molybdophosphoric acid followed by heating at 100 °C for 5 min. Phospholipids and aminolipids were detected by spraying TLC plates with the Dittmer-Lester and ninhydrin reagents, respectively, and glycolipids were visualized with α-naphtol [29].

### 2.4. Carotenoid Production

Flasks of 250 mL volume containing 80 mL of culture medium were inoculated with all isolates and incubated at 37 °C, 200 rpm, pH 8. Aliquots were taken every 24 h for 6 days to measure biomass and carotenoid production. Biomass was quantified using a correlation between the optical density (OD_620_) and dry weight: biomass g/L = 0.688 × OD_620_. The pigment extraction was performed using a modified Naziri method [30] that consisted firstly of obtaining the cellular pellets by centrifugation at 4000 rpm, 4 °C for 1 h, and frozen at −20 °C. Then, a solution of acetone/methanol (7:3, *v*/*v*) was added to the cellular pellets and the cell membranes were disrupted by vortexing and sonicating. Afterwards, the mixtures were centrifuged at 6000 rpm, 4 °C for 30 min, and the supernatants were collected. Finally, the solvent was evaporated using a SpeedVac concentrator (ThermoFisher Scientific, MA, USA).

The carotenoid quantification was calculated with the Lambert–Beer law as Britton describes it [31], reading absorbance at λ490 nm and using 2660 as the molar attenuation coefficient of bacterioruberin in methanol. The most carotenoid-producing strain of each genus was selected for further studies.

To identify the main pigment, the carotenoid extracts were suspended in acetone and a spectrophotometric scan from λ400 to λ600 nm was carried out in a Biotek Eon Microplate Spectrophotometer (Agilent, Santa Clara, CA, USA).

Experiments were carried out in triplicate.

### 2.5. Optimization of Carotenoid Production: Effect of Carbon Sources

The most carotenoid-producing strain of each genus was selected for deeper investigation; in particular, the microbial response to carbon source presence was evaluated. For this purpose, three different carbon sources, namely, glucose (10 and 25 g/L), succinic acid (3.5 g/L) or glutamic acid (3 g/L), were added at a time to 40 mL of the culture medium ATTCC2185 at the optimal conditions of each isolate. After regular interval times, growth was monitored for 120 h by measuring the OD_620_ for biomass calculation according to the equation described above and by extracting the carotenoids as reported. Experiments were carried out in triplicate.

### 2.6. Statistical Analysis

Statistical analysis was performed using a one-way analysis of variance (ANOVA) to test the significance of the model (S1). Significance was determined at *p* < 0.05.

## 3. Results

### 3.1. Sampling and Isolation

Santa Bárbara saltern (Figure 2) is a salt-producing area in a non-protected zone located in the Municipality of Huatabampo, Sonora, México, at 10 m above sea level. It belongs to the physiographic province of the pacific coastal plain. This region is a flat relief next to the ocean with flooded areas parallel to the coast covered by alluvial sediments from the Sierra Madre Occidental [32]. The weather is dry with an average temperature of 23.4 °C, ranging from 0 °C in winter to 49 °C in summer. The annual precipitation is less than 300 mm; the fauna is composed of a variety of endemic birds, reptiles, amphibians, felines and turtles; and the vegetation is mainly xerophytic and halophilic plants [33,34].

Seventeen strains were obtained from the sample isolation and eight of them, the most pigmented, were selected for further investigation. All isolates (Figure 3) are pleomorphic, Gram-negative and orange-red colored.

### 3.2. Identification of Halophilic Archaea Strains

The closest relative of each strain in the database, as found with BLAST, is showed in Table 1. To achieve an unambiguous taxonomic identification of the isolates, a phylogeny was reconstructed using 16S rRNA genes (Figure 4). All isolates belong to three genera *Haloarcula*, *Halorubrum* and *Halolamina*, and will be further identified with the strain code.

Neutral lipid identification (quinones) by LC–mass spectrometry (Table 2) showed menaquinone with eight isoprenoid units (MK-8; 717 g/mol molecular mass) for all strains except strain M5; in addition, strains M2, M3, M4, M6 and M7 exhibited dihydromenaquinones-8 (MK-8 (H2); 719 g/mol molecular mass). Finally, strain M5 showed methylmenaquinone-8 (MMK-8; 733 g/mol molecular mass).

The polar lipid profile of the strains comprised derivatives of the polar phospholipids (PG), in which the hydrocarbon isoprenoid structure is composed by two chains containing 20 carbons (diether C_20_-C_20_ moiety) linked by an ether linkage to the glycerol structure. Phosphatidylglycerol phosphate methyl ester (PGP-Me), sodium salt 1-(3-sn-phosphatidyl)-*rac*-glycerol (PI) and unknown glycolipids (GP) were found in all isolated strains; in addition, derivatives of unidentified glyco-phospholipid were found in strains M1, M3 and M4 and, finally, mannose-6-sulfate(1-2)-glucose glycerol diether (S-DGD) was described for strain M3. Two unidentified aminolipids were detected in all isolated strains.

### 3.3. Carotenoid Producers

All strains were carotenoid producers; the highest carotenoid production started from 136–160 h, at the stationary phase. A graph of the yield of carotenoids per gram of biomass produced by each strain is found in Figure 5.

The highest yield was achieved by the *Halorubrum* genus, especially in *Halorubrum* sp. M5 at 160 h with 30.55 mg/g of biomass.

#### 3.3.1. Carbon Source Influence in Carotenoid Production

Statistical analysis showed, in strain *Haloarcula* sp. M1, significant differences in growth when an extra carbon source was added to the medium with respect to the control medium (*p*-value = 0.001), but no differences were observed between the carbon source used; whereas, pigment production improved with both glucose concentrations employed, especially at 2.5% (*p*-value = 0.0000004). *Halolamina* sp M3 showed no differences comparing to control for growth (*p*-value = 0.322), and pigmentation was negatively affected when glucose was added to the control medium (*p*-value = 0.00000003). Glucose also stimulated growth (*p*-value = 0.0000000000004) and pigmentation (*p*-value = 0.00000002) in *Halorubrum* sp. M5, 1% glucose being the carbon source that improved the most pigment production, reaching 23.21 mg/g of biomass at 120 h. In *Halorubrum* sp. M5, pigment production was affected negatively by glutamic acid. The biomass and carotenoid production of the three selected strains are shown in Figure 6. Detailed statistical analysis is provided in Appendix A.

#### 3.3.2. Pigments Identification

The eight strains were able to produce carotenoids as revealed from spectrophotometric investigation. The spectrophotometric scans showed the three characteristic peaks of bacterioruberin in acetone (λ468, 498 and 532 nm) according to Britton’s carotenoid manual [31], which confirms the presence of bacterioruberin as the main pigment in all of the strains. The scan is shown in Figure 7. Although bacterioruberin is the main pigment, there may be other carotenoids in which its spectrum is hidden behind that of bacterioruberin, as chromatographic characterization is necessary to detect all pigments.

## 4. Discussion

Pigment-producing haloarchaea inhabit high salinity environments worldwide [39]. In this work, we successfully isolated eight haloarchaeal strains from the Santa Bárbara saltern in northwest México, which produce pigments of biotechnological potential. These isolates belong to three genera (*Haloarcula*, *Halolamina* and *Halorubrum*) previously found to be pigmented strains [40,41,42]; however, to our knowledge, this is the first report of an isolate of the genus *Halolamina* being evaluated on pigment production potential. Unlike other members of the *Halobacteriaceae* family, *Halolamina* has been recently isolated and recognized as a novel genus [43]. As observed in Figure 2, phylogenetic reconstruction shows that each isolate might represent different species; nevertheless, a genome-based taxonomy is currently directing research on uncultivated species [44,45] as culturing and sequencing techniques keep expanding the tree of life in the Archaea domain [46].

Menaquinone-8 (MK-8) is commonly found in halophilic archaea; its weight is 717 Da and 719 Da in its hydrogenated form [47]. Several studies report MK-8 in *Haloarcula* [48,49,50] and in *Halorubrum* [51,52], but so far, we have not found any reports of menaquinones in *Halolamina*. The weight 733 Da found in strain *Halorubrum* sp. M5 corresponds to methyl-menaquinone-8 (MMK-8) also found in *Natronobacterium gregoryi* [53].

The glycolipid S-DGD could be associated with gas vesicles [54]; it was reported by Cui when proposing the genus *Halolamina* [43] and it is described in other *Halolamina* species [41]. In this work, it was found only in *Halolamina* sp. M3. In the *Haloarcula* genus it has not been reported yet. PG and PGP-Me were present in all strains; this lipid is related with cytochrome C oxidase and alkalophilic archaea lack this phospholipid [55].

The strains have the capacity to grow at various values of NaCl, pH and all temperature; this may be due to the ever-changing ecosystem from which the archaea were isolated, where temperature and water retention varies depending on the season [10]. This represents a great advantage for industrial culturing, since it reduces incubation and cooling costs.

The maximum carotenoid yield of 30.55 mg/g obtained in *Halorubrum* sp. M*5* is similar than those obtained in optimized conditions by de la Vega [42] in the same genus, species *Halorubrum SH1*. This strain was selected to investigate the effect of carbon source on pigment production due to its high pigmentation, whereas *Haloarcula* sp. M1 and *Halolamina* sp. M3 were chosen for being the only ones of their genus in our study.

Most of the studies regarding haloarchaea pigment production are focused on culture conditions, such as the salt concentration, agitation, light or oxygen content [5,40,42]. However, the research related to the effect of carbon sources in pigment production by haloarchaea is scarce, although glucose and glycerol have been studied on *Halobacterium* [5,21]. While succinic and glutamic acid had been reported as carbon sources for some archaea, their effect on pigment production has not been elucidated. Regarding the archaeal metabolism, not all haloarchaea function with the same metabolism, as the utilization of metabolic pathways depends not entirely on the genetics but also on ecology.

Glutamic acid assimilation in halophilic archaea has not been fully elucidated. *Halobacterium* contain enzymes that allow glutamic acid to be directly incorporated into the tricarboxylic acid cycle (TCA) and obtain acetyl-CoA [17]; it can be metabolized by *Halobacterium salinarum* and *Haloarcula marismortui* [56,57]. However, other authors report that *Haloarcula* and *Halorubrum* are incapable to grow in glutamic acid [14], but *Haloarcula* metabolizes glutamic acid when it is used as the sole carbon source [58]. Our results indicate that addition of glutamic acid in the control medium did not increase the growth on *Halolamina* sp. M3, and a negative effect on growth was observed for *Halorubrum* sp. M5 whereas a slight increment on growth was achieved for *Haloarcula* sp. M1. Besides, pigment production was not affected in *Haloarcula* sp. M1 and *Halolamina* sp. M3 but negatively affected *Halorubrum* sp. M5. It could mean that glutamic acid affects growth unless it is the only carbon source available; that the ability to metabolize it varies in species within the same genus; and that it does not have a positive effect on pigment production.

The addition of succinic acid in the control medium was proposed on the basis that halophilic archaea contain enzymes that allow amino acids to be directly incorporated into the tricarboxylic acid cycle [17]. The effect of succinic acid in carotenoid production in yeasts is remarkable, as reported for *Xantophyllomyces dendrorhous* [18]. The effect of succinic acid on the growth of the strains studied was similar to those observed with glutamic acid; however, succinic acid did not increase the pigment production of the three strains studied, although halophilic archaea and yeasts possess similar pathways and enzymes. Ultimately, they are phylogenetically further apart, and more studies about the impact of amino acids on archaeal pigment production are needed.

Glucose metabolism is complex and variable in halophiles; it is degraded via different variants of the ED pathway within them [57]. In the most known way, glucose is converted to pyruvate via ED, a carboxyl group is removed, and coenzyme A is attached to form acetyl-CoA [19]. The effect of glucose on growth and pigmentation was evaluated in *Halobacterium* by Goshnauer, obtaining inhibition of pigmentation when 4% glucose was used as carbon source, and little effect on growth but considerable improvement on pigmentation with 2% of glucose [21]. Our results suggest that the effect of glucose on pigment production depends on the species. In *Haloarcula* sp. M1 and *Halorubrum* sp. M5, pigment production improved when glucose was added, but in *Halolamina* sp. M3 pigment production decreased. The effect of glucose on growth was evident in *Halorubrum* sp. M5 and *Haloarcula* sp. M1 but not in *Halolamina* sp. M3. Glucose concentration has different effects, depending on the strain. In the case of *Haloarcula* sp. M1, 2.5% glucose increased pigment production, where as in *Halorubrum* spp. pigment production increased with 1% glucose. Finally pigment production can be improved using single carbon sources as glucose, but not for all halophilic archaea. A deeper understanding of pigments metabolism in haloarchaea is needed, as well as research on pigment production, combining carbon sources with growth parameters such as oxygen, light or salt concentration.

## 5. Conclusions

This work represents first evidence of halophilic archaea in the Santa Barbara saltern, exhibiting at least eight species distributed across three different genera of halophilic archaea, namely, *Haloarcula*, *Halolamina* and *Halorubrum*, with the possibility that *Halolamina* sp. strain M3 could be a new species. All eight strains are carotenoid producers with bacterioruberin as the main pigment. There is a relationship between growth and pigment production, although the latter can also be influenced by stress factors and components in the culture media. The addition of glucose to the ATCC2185 media increases considerably the production of carotenoids and growth in *Haloarcula* sp. M1 and *Halorubrum* sp. M5 but not in *Halolamina* sp. M3. This paper highlights the diversity of carbohydrate metabolism in haloarchaea and their effect on carotenoid production.

## Figures and Tables

**Figure 1 microorganisms-09-01096-f001:**
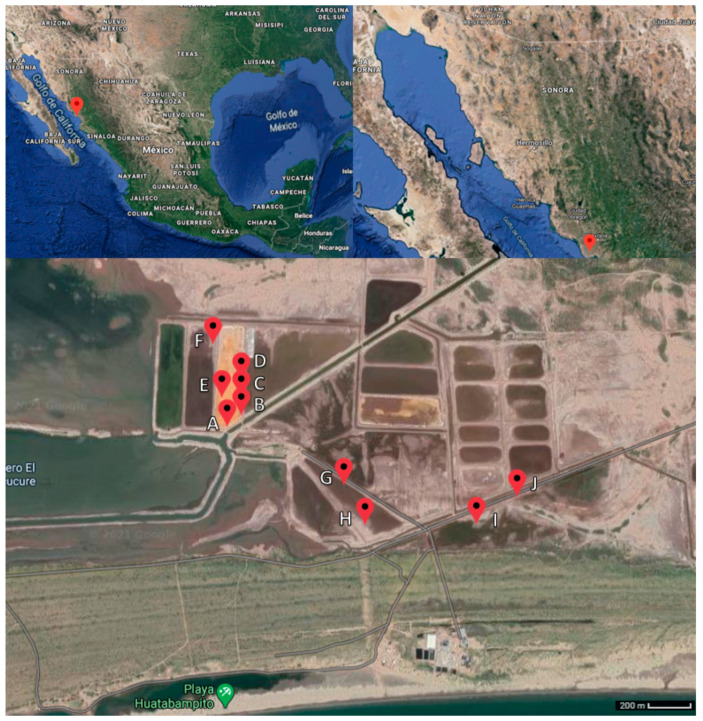
Maps of Huatabampo municipality in México, in the Sonora State, and of the Santa Bárbara saltern, showing the 10 sampling points (A: N26°42.421′W109°38.510′; B: N26°42.444′W109°38.475′; C: N26°42.469′W109°38.468′; D: N26°42.512′W109°38.469′; E: N26°42.498′W109°38.530′; F: N26°42.619′W109°38.554′; G: N26°42.269′W109°38.176′; H: N26°42.175′W109°38.119′; I: N26°42.167′W109°37.800′; J: N26°42.240′W109°37.677′). Satellite photo from Google maps.

**Figure 2 microorganisms-09-01096-f002:**
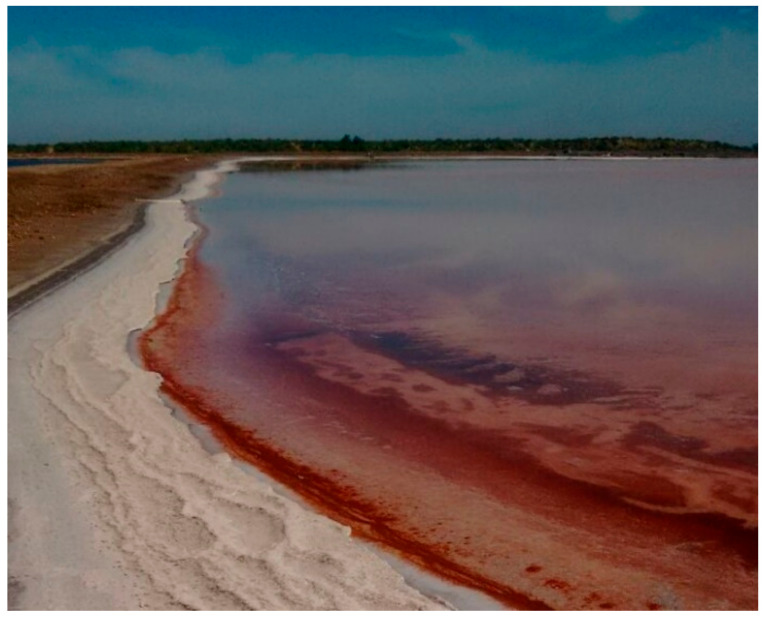
Santa Bárbara saltern, Huatabampo, Sonora.

**Figure 3 microorganisms-09-01096-f003:**
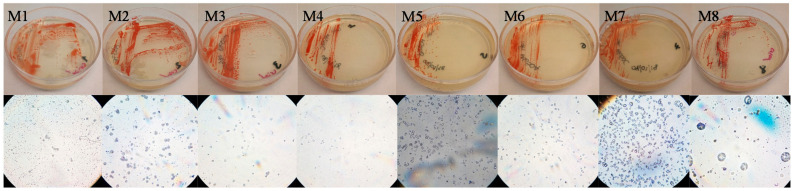
Petri and microscope pictures of the eight isolates.

**Figure 4 microorganisms-09-01096-f004:**
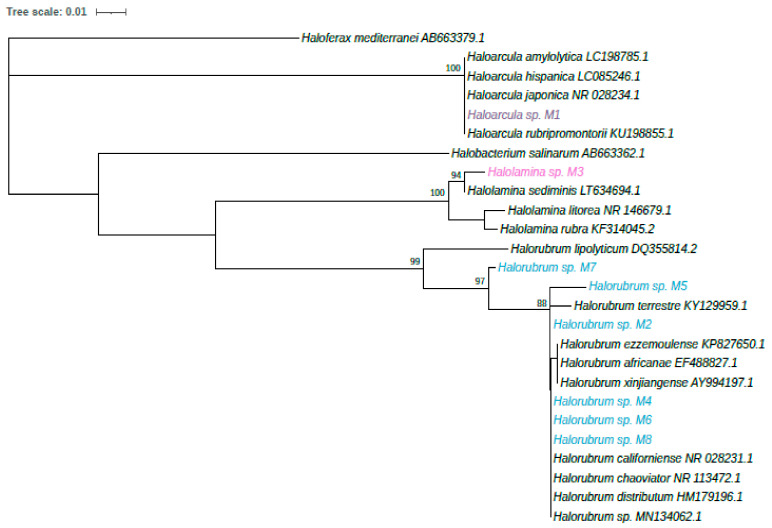
Maximum likelihood phylogenetic tree showing the relationship between the isolated strains and related taxa. Reconstruction was based on partial 16S rRNA sequences with the calculated T92+1 evolutionary model. Bootstrap values, indicated at nodes harboring the studied strains, were obtained from 1000 bootstrap replicates and are reported as percentages. *Haloferax mediterranei* was used as the outgroup. The closest reference strains are shown in black.

**Figure 5 microorganisms-09-01096-f005:**
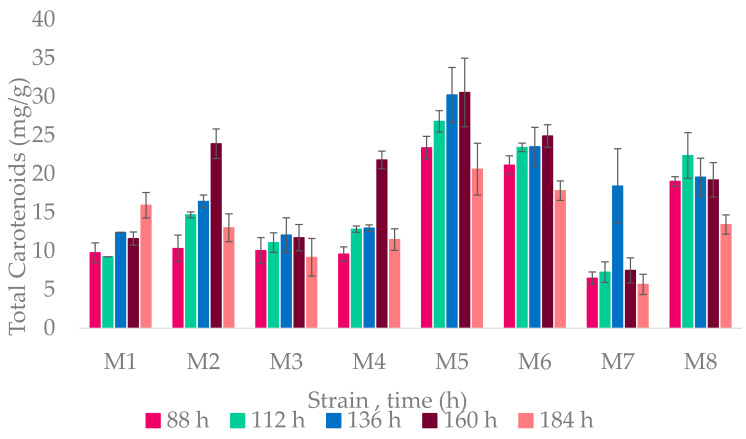
Yield of pigments produced per gram of dry weight biomass. On the X axis, the time in hours and the strain code; on the Y axis, the total yield of carotenoids expressed in mg/g.

**Figure 6 microorganisms-09-01096-f006:**
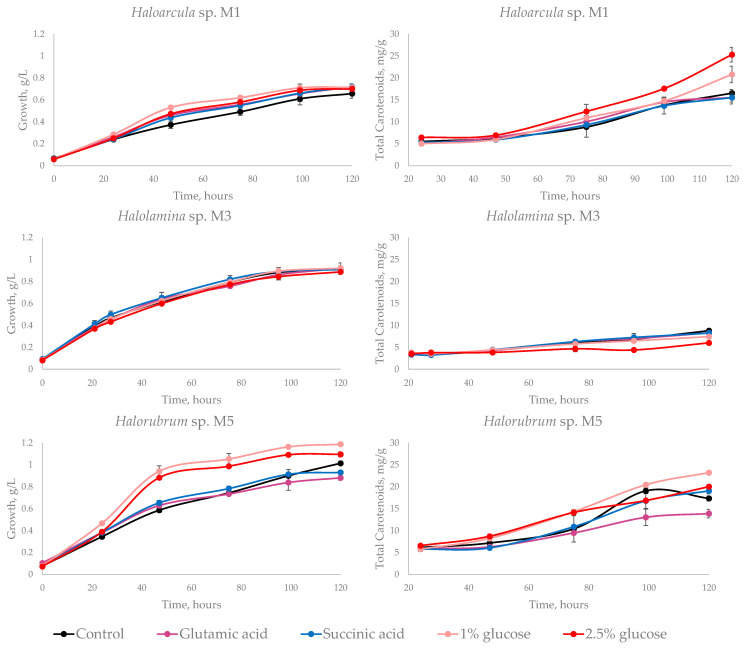
Effect of carbon source on growth and carotenoid production; the standard deviation at each point is shown.

**Figure 7 microorganisms-09-01096-f007:**
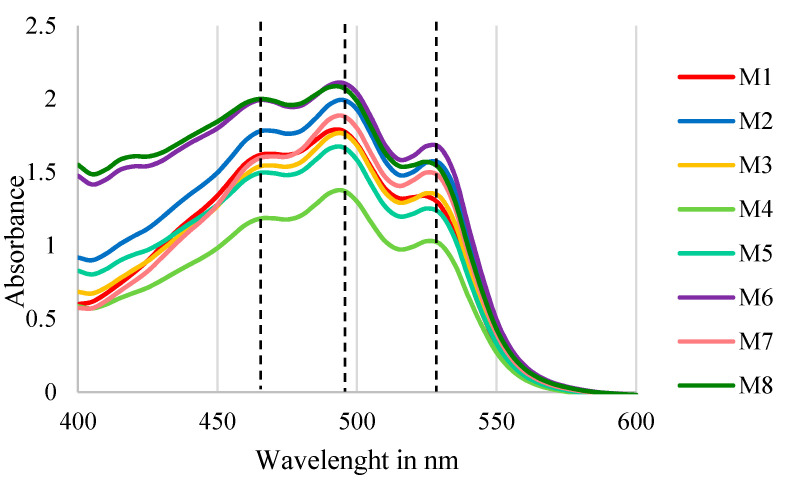
Spectrophotometric scans of the pigment extracts from all isolated strains.

**Table 1 microorganisms-09-01096-t001:** Closest related strains of isolates based on 16S rRNA gene sequence similarity.

Strains Code	Closest Strains in GenBank Database	Accession Numbers	Percentage of Similarity	Reference
M1 CA_13B53	*Haloarcula* sp. M1 *Haloarcula salaria*	MW567147 LN649977.1	100% 99.4%	From this study [35]
M2 Fb21	*Halorubrum* sp. M2 *Halorubrum ezzemoulense*	MW567148 CP034940.1	100% 99.3%	From this study [36]
M3 UAH-SP14	*Halolamina* sp. M3 *Haloarchaeon UA-SP14*	MW567149 HM031393.1	100% 99.46%	From this study [37]
M4 SD683	*Halorubrum* sp. M4 *Halorubrum* sp. *SD683*	MW567151 LT578362.2	100% 99.85%	From this study [38]
M5 Fb21	*Halorubrum* sp. M5 *Halorubrum ezzemoulense*	MW567150 CP034940.1	100% 93.3%	From this study [36]
M6 Fb21	*Halorubrum* sp. M6 *Halorubrum ezzemoulense*	MW567152 CP034940.1	100% 99.85%	From this study [36]
M7 Fb21	*Halorubrum* sp. M7 *Halorubrum ezzemoulense*	MW56567153 CP034940.1	100% 100%	From this study [36]
M8 E302-1	*Halorubrum* sp. M8 *Halorubrum* sp. *E302-1*	MW567154 JN196504.1	100% 99.17%	From this study [38]

**Table 2 microorganisms-09-01096-t002:** The eight strains and their lipid content, MK-8 = menaquinone-8; UK = unknown lipid; GP = unidentified glycol-phospholipid; PG = polar phospholipids; PGP-Me = methylated polar phospholipids; PI-Na = phosphatidylinositol sodium salt; S-DGD = mannose-6-sulfate(1-2)-glucose glycerol diether.

Strain ID	Quinone	Glycolipids	Phospholipids
*Haloarcula* sp. M1	MK-8	2 UK, 1 GP	PG, PGP-Me, PI-Na, 3 UK
*Halorubrum* sp. M2	MK-8, MK-8(H_2_)	1 UK	PG, PGP-Me, PI-Na, 3 UK
*Halolamina* sp. M3	MK-8, MK-8(H_2_)	S-DGD, 1 UK, 1 GP	PG, PGP-Me, PI-Na, 2 UK
*Halorubrum* sp. M4	MK-8, MK-8(H_2_)	2 UK, 1 GP	PG, PGP-Me, PI-Na, 3 UK
*Halorubrum* sp. M5	MMK-8	1 UK	PG, PGP-Me, PI-Na, 3 UK
*Halorubrum* sp. M6	MK-8, MK-8(H_2_)	2 UK	PG, PGP-Me, PI-Na, 3 UK
*Halorubrum* sp. M7	MK-8, MK-8(H_2_)	1 UK	PG, PGP-Me, PI-Na, 3 UK
*Halorubrum* sp. M8	MK-8	1 UK	PG, PGP-Me, PI-Na, 3 UK

## Data Availability

The data presented in this study are available on request from the corresponding author. The data are not publicly available due to data are in process of publishing in the CIATEJ public repository as part of Sofia Vazquez thesis.

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
