# Peer review of "Effect of Carbon Sources in Carotenoid Production from Haloarcula sp. M1, Halolamina sp. M3 and Halorubrum sp. M5, Halophilic Archaea Isolated from Sonora Saltern, Mexico"

_microorganisms, 2021, doi:10.3390/microorganisms9051096_

Round 1

Reviewer 1 Report

In the present work, Ana Sófia Vásquez-Madrigal et al. for the first time show the presence of halophilic archaea in Santa Bárbara saltern, México. In addition, they deeply inspect their carotenoid production and reveal that it is dependent on carbon sources. The study is very well written and a lot of complementary approaches (bioinformatics, PCR, sequencing, LC/MS, NMR, UV spectroscopy) were used. Nonetheless, there are some issues that should be fixed prior to publication.

Major points:

  • chapter 3.3.1 and Figure 6 ... statistics are missing here, it is necessary to enclose particular p-values (e.g. to the supplementary material, or to the main text, where appropriate)
  • section 2.1 ... please, add detailed cultivation conditions (light, oxygen)
  • section 2.2 ...PCR conditions missing (temperatures, times, no. of cycles) - this would be important information for repeatability of the experiment + describe details of the sequencing procedure
  • some critical BLASTn parameters missing (database, non-redundant (?); organism, limited to Archaea domain (?)) + T92+1 model (how it was selected (?)
  • The author's contribution section is missing

Minor points:

line 32-33 ... the sentence is wordy, not sure what you want to say, sounds rather like a common phrase

line 45 ... are carotenoid pigments exclusive to halophilic archaea or found in Bacteria also? Just interested (and readers also will), please, put it to the broader context

line 53 ... Santa Bárbara ... where does the salinity come from? Should be explained to the readers. Besides, is it a protected area/national park?

line 107 ... generally employed ... some reference needed

Figure 3 ... image quality seems to be very low in my PDF

Figure 4 ... some bootstrap values are missing

lines 214 - 216 ... unclear sentence to me ... all cases 717 Da vs. M3, M4, M6, M7, M8 719 Da, and M5 733 Da (?)

line 217 ... length 20C-20C (?) ...seems like a nonsense range to me

line 225 ... Weights 717 and 719 found in all isolates ?!? This is not consistent with the previous paragraph (lines 214-216), please, double-check the whole section

Figure 7 ... would be helpful to highlight 3 characteristics peaks by arrows or dashed vertical lines

line 323-326 ... the sentence is wordy

line 324 and 338 ... specie (it is not right) ... replace by "species"

line 343 ... the importance of deep understanding is revealed (?)

Reference section ... journal names should be unified ... full names vs. abbreviations

Author Response

We are grateful for your suggestions, all your comments were attended, changes were highlighted in yellow in the manuscript. Responses are detailed in the submited document. 

Reviewer 2 Report

In this work, the authors describe the effect of different carbon sources in carotenoid production by a new archea isolated from Sonora Saltern in Mexico.

The work is well written, and experimental data support all conclusions drawn.

There are only a few things to fix in the materials and methods, in particular, it would be interesting to know the temperature and pH of the mud where the sampling was carried out.

Another very minor problem is the description of the culture medium used for isolation. I think a more precise definition would make it more usable to the reader.

Author Response

We are grateful for your suggestions, all your comments were attended, changes were highlighted in yellow in the manuscript. Responses are detailed in the attached document.
